# Predicting a failure of postoperative thromboprophylaxis in non-small cell lung cancer: A stacking machine learning approach

**Ligang Hao[1]◉, Junjie Zhang[2]◉, Yonghui Di◉[1]\*, Zheng Qi[3], Peng Zhang[1]**

1 Department of Thoracic Surgery, Xingtai People's Hospital, Xingtai, Hebei, China, 2 Department of Computed Tomography and Magnetic Resonance, Xingtai People's Hospital, Xingtai, Hebei, China, 3 Department of Clinical Lab, Xingtai People's Hospital, Xingtai, Hebei, China

◉ These authors contributed equally to this work.
\* 1925965096@qq.com

## Abstract

### Background

Non-small-cell lung cancer (NSCLC) and its surgery significantly increase the venous thromboembolism (VTE) risk. This study explored the VTE risk factors and established a machine-learning model to predict a failure of postoperative thromboprophylaxis.

### Methods

This retrospective study included patients with NSCLC who underwent surgery between January 2018 and November 2022. The patients were randomized 7:3 to the training and test sets. Nine machine learning models were constructed. The three most predictive machine-learning classifiers were chosen as the first layer of the stacking machine-learning model, and logistic regression was the second layer of the meta-learning model.

### Results

This study included 362 patients, including 58 (16.0%) with VTE. Based on the multi-variable logistic regression analysis, age, platelets, D-dimers, albumin, smoking history, and epidermal growth factor receptor (EGFR) exon 21 mutation were used to develop the nine machine-learning models. LGBM Classifier, RandomForest Classifier, and GNB were chosen for the first layer of the stacking machine learning model. The area under the received operating characteristics curve (ROC-AUC), accuracy, sensitivity, and specificity of the stacking machine learning model in the training/test set were 0.984/0.979, 0.949/0.954, 0.935/1.000, and 0.958/0.887, respectively. In the validation set, the final stacking machine learning model demonstrated an ROC AUC of 0.983, accuracy of 0.937, sensitivity of 0.978, and specificity of 0.947. The decision curve analyses revealed high benefits.

**Data availability statement:** All data generated or analyzed during this study are included in this article.

**Funding:** This work was supported by the Key Development Plan of XingTai (ZC20301 to Junjie Zhang and 2022ZC271 to Ligang Hao) in study design, data collection and analysis, decision to publish and preparation of the manuscript.

**Competing interests:** The authors have declared that no competing interests exist.

## Conclusion

The stacking machine learning model based on EGFR mutation and clinical characteristics had a predictive value for postoperative VTE in patients with NSCLC.

## Introduction

Venous thromboembolism (VTE) is defined as pulmonary embolism, deep venous thrombosis, abdominal venous thrombosis (e.g., liver or splanchnic venous thrombosis), or other types of venous thromboembolism [1]. Many diseases increase the risk of VTE, especially cancer [2]. The annual incidence of VTE in patients with cancer is 0.5%–20%, depending upon the cancer type and other risk factors, and reaches 70% in selected populations [3]. VTE is one of the leading causes of death in cancer patients undergoing surgery or receiving radiochemotherapy [2]. Nevertheless, predicting who will suffer from VTE remains difficult.

Lung cancer is the second most common cancer worldwide [4], and non-small cell lung cancer (NSCLC) represents 85%–90% of all lung cancers [5]. NSCLC has a high rate of complication with VTE, especially after surgery [6,7]. Lung cancer is the most commonly identified malignancy in patients with VTE, with an incidence of 3%–13.9% in patients with VTE and 3.8% in patients with pulmonary embolism (PE) [8]. Surgery significantly increases the risk of postoperative VTE and PE-related death by 2 and 4 folds, respectively [9]. Still, VTE and PE are often asymptomatic or present insidiously with only nonspecific symptoms [10,11]. Therefore, establishing a method to predict VTE and help make precision preventive strategies is urgently needed.

In daily clinical practice, a variety of risk assessment models for VTE were used, including the Caprini score system [12], Rogers score system [13], Padua score system [14], and Khorana score system [15]. In recent years, the modified Caprini score has been used to assess the risk of VTE in patients undergoing thoracic surgery [16]. Still, all these models were established using Western mostly-Caucasian-based patient populations and/or not based on patients in the thoracic surgery department. Indeed, the area under the curve (AUC) of the modified Caprini score is only 0.474 in Chinese patients with pulmonary surgery, with few patients in the high-risk group and a poor relationship between the risk of VTE and the actual VTE occurrence [17]. A previous study showed that age, duration of operation, lymphocyte count, platelet count, and D-dimer levels were independent predictive factors in patients with lung cancer [18], but the lymphocyte count, platelet count, and D-dimer levels are not included in the Caprini score system. A model developed from Chinese patients showed an AUC of 0.80 [18]. Furthermore, the biology of the underlying cancer can influence the occurrence of VTE, especially in patients with ROS proto-oncogene 1 (ROS1) fusion, anaplastic lymphoma kinase (ALK) fusion, and Kirsten rat sarcoma viral oncogene (KRAS) mutation [19,20]. Nevertheless, the correlation between VTE and epidermal growth factor receptor (EGFR) mutations, the most common driver mutation type in Asians, remains highly controversial [20–23]. Until now, no model has explored the impact of NSCLC driver mutation on VTE risk after surgery.

Therefore, this study explored the correlation between EGFR mutation status and VTE and used machine learning algorithms to determine the optimal combination of biomarkers to predict the risk of VTE in patients with NSCLC undergoing surgery and postoperative thromboprophylaxis. The results could help predict VTE in patients with NSCLC after surgery and improve patient management and prognosis.

## MATERIALS AND METHODS

### Study design and patients

This retrospective study included patients with non-small-cell lung adenocarcinoma who underwent surgery between January 2018 and November 2022 in the Xingtai People's Hospital. And all data collection and analysis of these patients was done between November 2022 and February 2023. This work has been carried out in accordance with the Declaration of Helsinki (2000) of the World Medical Association. This study was approved by the Ethics Committee of Xingtai People's Hospital (# 2022[012]). Given the nature of the retrospective study, the requirement for individual informed consent was waived by the Ethics Committee of Xingtai People's Hospital. Authors had no access to information that could identify individual participants during or after data collection.

The inclusion criteria were 1) patients with NSCLC diagnosed by pathological examination of the surgical specimen, 2) complete clinical data, including VTE data and EGFR mutation status, and 3) received systemic VTE prophylaxis. The exclusion criteria were 1) neoadjuvant therapy (including radiotherapy, chemotherapy, chemoradiotherapy, or molecular targeted therapy) or 2) a history of VTE before surgery.

All patients enrolled were received lobe resection and systemic lymph node resection. Nadroparin was administered subcutaneously for a total of 28 days after surgery to prevent thromboembolism. Patients underwent weekly screening for deep venous thrombosis (DVT) using lower-extremity ultrasonography for the first 4 weeks after surgery and then as clinically needed.

### Data collection and definition

The data extracted from the medical records of the patients included age, sex, smoking status, clinical stage, EGFR mutation status (wild type and exon 19, exon 20, and exon 21 mutations), blood cell counts (white blood cell count, red blood cell count, and platelet (PLT) count), and coagulation parameters (D-dimer, fibrinogen, and activated partial thromboplastin time (APTT)), albumin, and fasting blood glucose within 1 week before surgery. Lower-extremity ultrasonography was used to screen for deep venous thrombosis (DVT) before surgery, 3 days after surgery, and at any time when suspecting the possibility of DVT. Computed tomography angiography (CTA) of the pulmonary artery was used to screen for pulmonary embolism (PE) in patients suspected of PE during postoperative hospitalization. The Caprini score [12] was used to guide the VTE prophylaxis.

### Statistical analysis

All statistical analyses were performed using Python version 3.7. Continuous data with a normal distribution were presented as means ± standard deviations (SD) and analyzed using the independent-samples t-test. Continuous data with a skew distribution were presented as median (interquartile range, IQR) and analyzed using the Mann-Whitney U-test. The categorical data were presented as n (%) and analyzed using the chi-square test or Fisher's exact test. The factors with $P < 0.10$ were included in the logistic regression multivariable analysis.

The patients were randomly split into the training and test sets in a 7:3 ratio. Then, based on the factors with $P < 0.05$ in the multivariable analysis in the training set, nine machine learning models were developed using the training set: XGB Classifier, LGBM Classifier, 'RandomForest Classifier, AdaBoost Classifier, GaussianNB, LogisticRegression, MLP Classifier, SVC, and KNeighbors Classifier. The optimal parameters in the nine models were retrospectively identified using 5-fold cross-validation. The XGB Classifier was

implemented using XGBoost1.2.1. The LGBM Classifier was implemented using lightgbm 3.2.1. The others were implemented using sklearn 0.22.1. The receiver operating characteristic (ROC) curve was used to evaluate the performance of the nine machine-learning models based on the AUC. A 5-fold cross-validation was used for the validation of the best efficient machine-learning model. The main evaluation indicators were ROC AUC, accuracy, sensitivity, specificity, and positive and negative predictive values. According to the ROC, the three most predictive machine-learning classifiers were chosen as the first layer of the stacking machine-learning model, and logistic regression was chosen as the second layer meta-learning model [24]. For the stacking machine learning model, in the training set, 10-fold staking required dividing 10 datasets, meaning that each base learner needed to iterate 10 times. In the test set, since each base learner previously trained a model for each fold (that is, one base learner actually trained 10 models), these 10 models made overall predictions for the test set, and the results were obtained, and the score of VTE was calculated (Fig 1). The score of the stacking machine learning model was represented using histograms in the training and test sets. Decision curve analysis (DCA) was used to calculate the clinical impact of the stacking machine learning model.

## RESULTS

### Patient characteristics

This retrospective study included 362 patients, after the 4-weeks follow-up, 58 (16.0%) were diagnosed with deep vein thrombosis of the lower extremity without any pulmonary embolism (PE). And the age ranged from 22 to 80. All of the patients enrolled were adenocarcinoma, for we go on EGFR mutation testing only in patients with adenocarcinoma after surgery. Patients were randomly divided into training group and test group at a ratio of 7:3, including 253 patients in the training group (207 patients without VTE 46 patients with VTE), and 109 patients in the test group (97 patients without VTE 12 patients with VTE). In all enrolled patients, there was no statistical difference in smoking history, gender, clinical stage, level of Fibrinogen, level of APTT and level of GLU between the two groups. And the average level of D- Dimers in patients with VTE was significantly higher than patients without VTE (0.55 vs 0.13; P < 0.001). While mean level of PLT and albumin was in patients with VTE was significantly higher than patients without VTE (221 vs 250, P < 0.001; 39.9 vs 42.5, P < 0.001). (Table 1)

### Correlation between VTE and EGFR mutation status

In all patients, 37.29% (135 patients) had EGFR mutation, including 76 patients with the exon 21 mutation and 59 with the exon 12 deletion. There were no other rare mutations. The patients with EGFR mutation had a higher ratio of VTE (22.2% vs. 12.33%, P = 0.013). The patients with the exon 19 mutation had a lower risk of VTE than those with the exon 21 mutation and had a similar risk compared with the patients with wild-type EGFR (Fig 2).

### Feature selection for model construction

In the training set, the multivariable analysis showed that patients with older age (OR = 1.109, 95%CI 1.055–1.146), elevated D-dimer levels (OR = 2.929, 95%CI: 1.677–5.714, P = 0.001), and EGFR exon 21 mutation (OR = 2.869, 95%CI: 1.352–6.062, P = 0.006) had a higher risk of VTE. On the other hand, the patients with elevated PLT (OR = 0.995, 95%CI: 0.990–0.999, P = 0.017), elevated albumin (OR = 0.922, 95%CI: 0.848–0.997, P = 0.051), and smoking history (OR = 0.404, 95%CI: 0.190–0.823, P = 0.015) had a lower risk of VTE (Table 2).

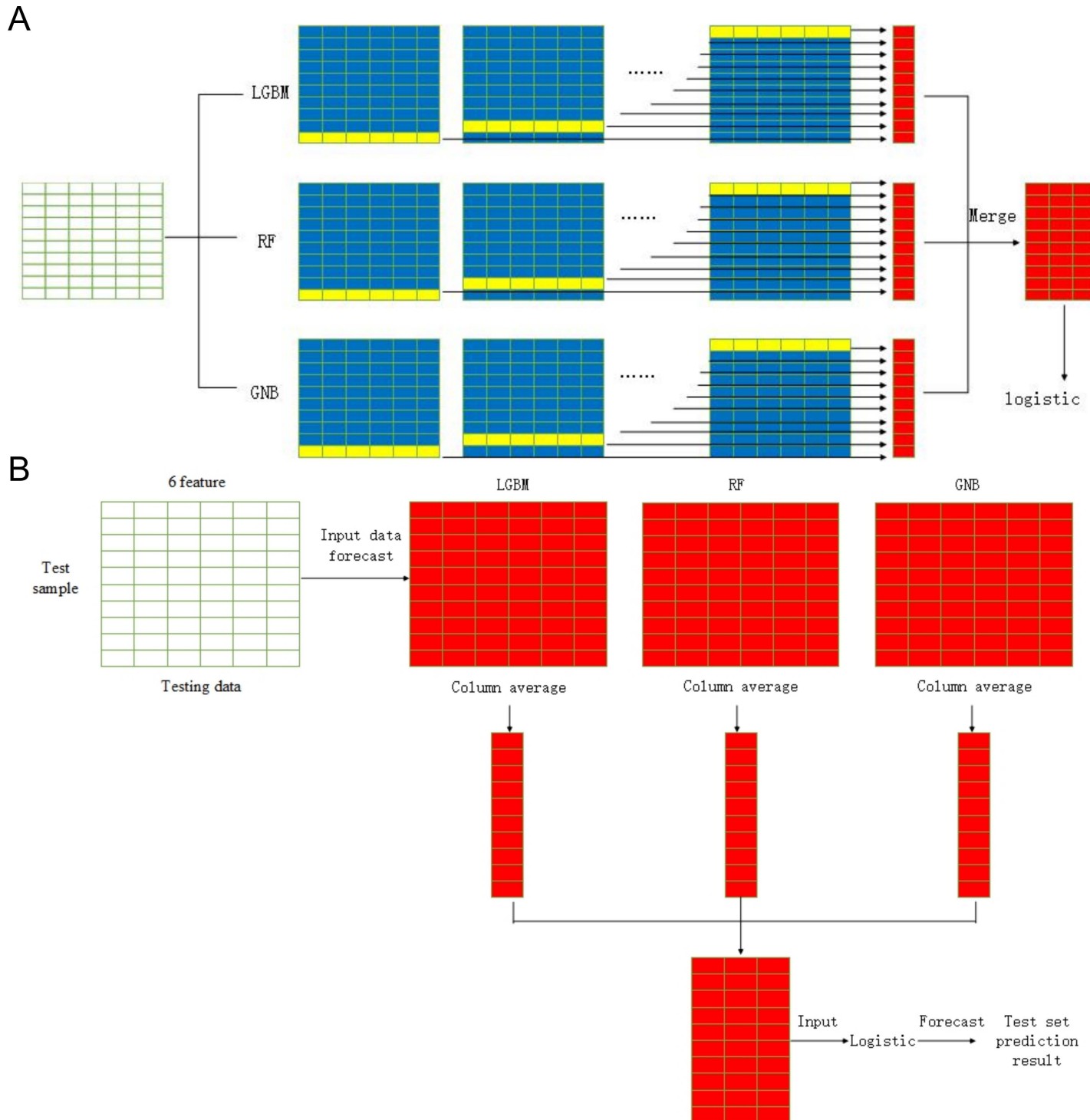

**Fig 1. Procedure of the stacking machine learning model to get the prediction results. (A)** In the training set, 10-fold staking requires dividing 10 datasets, meaning each base learner must iterate 10 times. **(B)** In the testing set, since each base learner previously trained a model for each fold (that is, one base learner actually trained 10 models), these 10 models made overall predictions for the test set, and the results were obtained.

**Table 1. Clinical characteristics of the patients.**

| Characteristics | All (n = 362) | Non-VTE (n = 304) | VTE (n = 58) | P |
|---|---|---|---|---|
| Sex | | | | 0.191 |
| Male | 203 (56.08) | 175 (57.57) | 28 (48.28) | |
| Female | 159 (43.92) | 129 (42.43) | 30 (51.72) | |
| Age (years) | 63 [55,68] | 62 [55,67] | 67 [64,72] | <0.001 |
| mEGFR | | | | 0.013 |
| No | 227 (62.71) | 199 (65.46) | 28 (48.28) | |
| Yes | 135 (37.29) | 105 (34.54) | 30 (51.72) | |
| mEGFR21 | | | | 0.002 |
| No | 286 (79.01) | 249 (81.91) | 37 (63.79) | |
| Yes | 76 (20.99) | 55 (18.09) | 21 (36.21) | |
| Smoking | | | | 0.095 |
| No | 201 (55.53) | 163 (53.62) | 38 (65.52) | |
| Yes | 161 (44.48) | 141 (46.38) | 20 (34.48) | |
| Stage | | | | 0.308 |
| I | 138 (41.95) | 120 (43.80) | 18 (32.73) | |
| II | 28 (8.51) | 23 (8.39) | 5 (9.09) | |
| III | 163 (49.54) | 131 (47.81) | 32 (58.18) | |
| Platelets (×10⁹/L) | 248.00 [196.00,303.00] | 250.00 [206.00,307.00] | 221.000 [165.00,280.00] | 0.003 |
| D-dimers (µg/mL) | 0.150 [0.10,0.36] | 0.133 [0.09,0.24] | 0.550 [0.18,1.03] | <0.001 |
| Albumin (g/L) | 42.200 [39.20,44.80] | 42.500 [39.80,45.00] | 39.900 [36.80,43.10] | <0.001 |
| Glucose (mmol/L) | 5.62 ± 1.43 | 5.60 ± 1.43 | 5.733 ± 1.41 | 0.526 |
| Fibrinogen (g/L) | 3.49 ± 0.99 | 3.46 ± 0.93 | 3.626 ± 1.25 | 0.247 |
| APTT (s) | 30.57 ± 3.12 | 30.53 ± 2.99 | 30.762 ± 3.68 | 0.601 |

Data are shown as mean ± standard deviation, median [interquartile range], or n (%).

VTE, venous thromboembolism; mEGFR, mutant epidermal growth factor receptor; mEGFR21, mutant epidermal growth factor receptor 21 exon; APTT, activated partial thromboplastin clotting time.

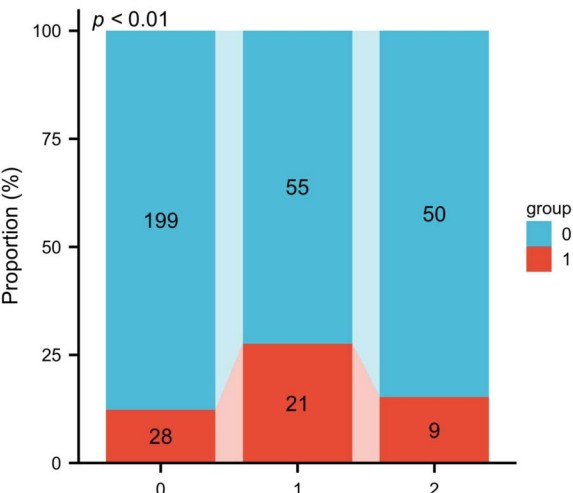

**Fig 2. Correlation between mutant forms of EGFR and venous thromboembolism (VTE).** X-axis represents mutant forms of EGFR: 0: wild type; 1: mutation in exon 21; 2: exon 19 deletion. Wild type and exon 19 mutation had a much lower risk of VTE than exon 21 mutation. Red bars indicate the patients with postoperative VTE, while blue bars indicate those without postoperative VTE.

**Table 2. Multivariable analysis to identify significant factors for VTE in the training set.**

| Characteristics | P | OR | 95%CI |
|---|---|---|---|
| Age | <0.001 | 1.097 | 1.055–1.146 |
| Platelets | 0.017 | 0.995 | 0.990–0.999 |
| D-dimers | 0.001 | 2.929 | 1.667–5.714 |
| Albumin | 0.051 | 0.922 | 0.848–0.997 |
| Smoking | 0.015 | 0.404 | 0.190–0.823 |
| mEGFR21 | 0.006 | 2.869 | 1.352–6.062 |

mEGFR21, mutant epidermal growth factor receptor 21 exon; OR, odds ratio.

## Construction of the stacking machine learning model

Based on the multivariable logistic regression analysis, age, PLT count, D-dimer levels, albumin levels, smoking history, and EGFR exon 21 mutation were used to develop the nine machine learning models. The largest ROC AUC averaged for each of the classifiers using 5-fold cross-validation in the training/test sets were 1.000/0.819 with the XGB Classifier, 1.000/0.959 with the LGBM Classifier, 1.000/0.895 with the RandomForest Classifier, 0.993/0.760 with the AdaBoost Classifier, 0.817/0.896 with the GaussianNB, 0.818/0.902 with LogisticRegression, 0.722/0.624 with the MLP Classifier, 0.819/0.761 with SVC, and 0.889/0.613 with the KNeighbors Classifier (Fig 3, Table 3, and Table 4).

The LGBM Classifier, RandomForest Classifier, and GaussianNB were chosen for the first layer of the stacking machine learning model, and logistic regression was chosen as the second layer meta-learning model. The VTE risk score for each patient in the training and test sets was calculated using the stacking machine learning model (Fig 4). ROC AUC, accuracy, sensitivity, and specificity of the stacking machine learning model in the training/test sets were 0.984/0.979, 0.949/0.954, 0.935/1.000, and 0.958/0.887, respectively (Fig 5A, C). In the validation test, the final model demonstrated an ROC AUC of 0.983, accuracy of 0.937, sensitivity of

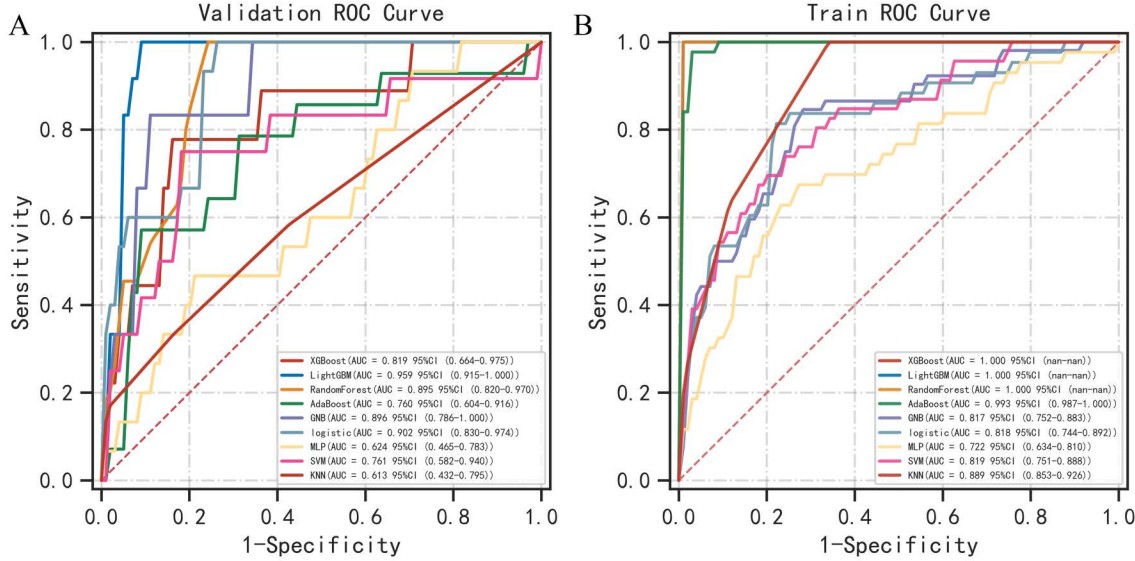

**Fig 3. Receiver operating characteristics (ROC) curves of the nine machine learning models in training and validation sets.**

**Table 3. Performance metrics for nine models in the training dataset.**

| Model | AUC | Accuracy | Sensitivity | Specificity | F1score |
|---|---|---|---|---|---|
| XGBoost | 1.000 | 0.997 | 1.000 | 1.000 | 1.000 |
| LightGBM | 1.000 | 0.997 | 1.000 | 1.000 | 1.000 |
| RandomForest | 1.000 | 0.993 | 1.000 | 1.000 | 1.000 |
| AdaBoost | 0.993 | 0.969 | 0.977 | 0.971 | 0.913 |
| GNB | 0.817 | 0.737 | 0.846 | 0.717 | 0.535 |
| Logistic | 0.818 | 0.782 | 0.814 | 0.780 | 0.524 |
| MLP | 0.722 | 0.723 | 0.674 | 0.736 | 0.416 |
| SVM | 0.819 | 0.754 | 0.739 | 0.765 | 0.484 |
| KNN | 0.889 | 0.844 | 1.000 | 0.658 | 0.674 |

AUC, area under the curve; XGBoost, EXtreme Gradient Boosting; SVM, polynomial supervised vector machine; LightGBM, Light Gradient Boosting Machine; AdaBoost, Adaptive boosting; GNB, Gaussian naive bayes; MLP, Multi-layer Perceptron; KNN, k-Nearest Neighbor.

**Table 4. Performance metrics for nine models in the validation dataset.**

| Model | AUC | Accuracy | Sensitivity | Specificity | F1score |
|---|---|---|---|---|---|
| XGBoost | 0.819 | 0.736 | 0.877 | 0.778 | 0.844 |
| LightGBM | 0.959 | 0.803 | 0.863 | 1.000 | 0.918 |
| RandomForest | 0.895 | 0.550 | 0.863 | 1.000 | 0.758 |
| AdaBoost | 0.760 | 0.495 | 0.808 | 0.571 | 0.915 |
| GNB | 0.896 | 0.053 | 0.671 | 0.833 | 0.896 |
| Logistic | 0.902 | 0.163 | 0.795 | 1.000 | 0.741 |
| MLP | 0.624 | 0.011 | 0.548 | 0.467 | 0.793 |
| SVM | 0.761 | 0.158 | 0.630 | 0.750 | 0.820 |
| KNN | 0.613 | 0.200 | 0.753 | 0.333 | 0.836 |

AUC, area under the curve; XGBoost, EXtreme Gradient Boosting; SVM, polynomial supervised vector machine; LightGBM, Light Gradient Boosting Machine; AdaBoost, Adaptive boosting; GNB, Gaussian naive bayes; MLP, Multi-layer Perceptron; KNN, k-Nearest Neighbor.

0.978, and specificity of 0.947 (Fig 5B). The training ROC AUC of the stacking machine learning model was very similar, ensuring that no model overfitting occurred. The DCA revealed that the net benefits of the stacking machine learning model for predicting postoperative VTE were very high (Fig 5D). The Brier Score of the predictive score of the stacking machine learning model was 0.032. The calibration plot is shown in Fig 5E. In the process of 5-fold cross-validation, the ROC AUC of the training and test sets were stable (Fig 5F).

## DISCUSSION

Age is a well-known risk factor for VTE [25]. Age is included in various models to predict the risk of VTE and guide the clinical strategy, such as the Padua, Caprini, and modified Caprini scores [12,14,17]. In the present study, age was independently associated with VTE, with an OR of 1.109, indicating that for every year of age, the incidence of VTE increases by 1.109 times. In our present study, the VTE rate of 16% after surgery in a group of patients who received 4 weeks of nadroparin prophylaxis is reported, slightly higher than previous study with a rate of 15.5%. It may be resulted from that much more stage III patients was enrolled in our study than previous study which also elevated the incidence of VTE.

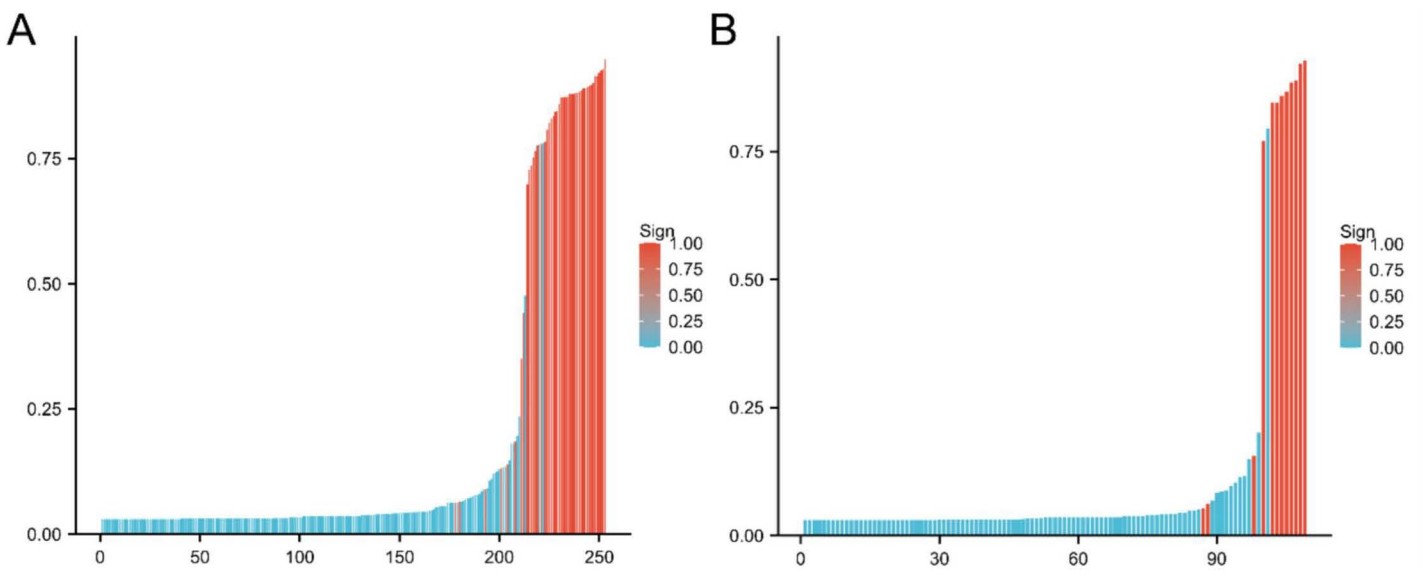

**Fig 4. Bar charts of the risk score for each patient in the training cohort.** (**A**) and testing cohort (**B**). The X-axis represents each patient. Each bar represents one patient. Red bars indicate the risk score for patients with VTE, while blue bars indicate the risk score for patients without VTE.

Elevated D-dimer levels reflect a hypercoagulable state [26,27]. The D-dimer levels are influenced by many factors, including surgery, infections, and anticoagulation [17,28]. In this study, the D-dimer levels were evaluated before surgery to reduce the influence of other factors as much as possible, so that it could reflect the hypercoagulable state of patients much more precisive. In this study, elevated D-dimer levels was also significantly related to VTE as an independent risk factor with OR = 2.929 which was familiar to previous study.

PLT aggregability and thrombocyte activation contribute to the pathogenesis of VTE [29], but the predictive value of PLT count for the risk of VTE is controversial [30]. In a meta-analysis, 16 studies were included, and six of them showed that lower PLT was associated with VTE, two studies showed that higher PLT was associated with VTE, and eight studies showed no differences in PLT between patients and controls [17]. The pooled analysis showed that the PLT values for patients were lower compared to controls [30]. Of note, all studies included in the previous meta-analysis were general patients [30]. In the present study, the results showed that the PLT counts in patients with lung cancer with VTE were significantly lower than in patients without VTE, with an of HR 0.995, meaning that for every $1 \times 10^9$ increase in PLT count decreased the VTE by 1.005 times.

The albumin levels and smoking history were negatively associated with VTE in the present study. Patients with VTE have lower albumin levels compared with non-VTE controls in patients with chronic liver disease or kidney disease [31,32]. Another study also revealed that serum albumin was an independent VTE risk factor in patients with cancer [33], supporting the present study. On the other hand, the influence of smoking on VTE in the study differed from the literature. Indeed, a study revealed that smoking increased the risk of VTE [34]. The risk of developing VTE was greater for current smokers than for former smokers, and a dose-response relationship was found for daily smoking and pack-years smoked [35]. The different outcome of this study may be because the pathology of the patients was adenocarcinoma. In such patients, ROS1 and ALK mutations are more common in patients without smoking [36]. A previous study showed that patients with ROS1 rearrangements had the highest incidence of VTE [20]. ALK rearrangements also increased the VTE risk [20]. In the

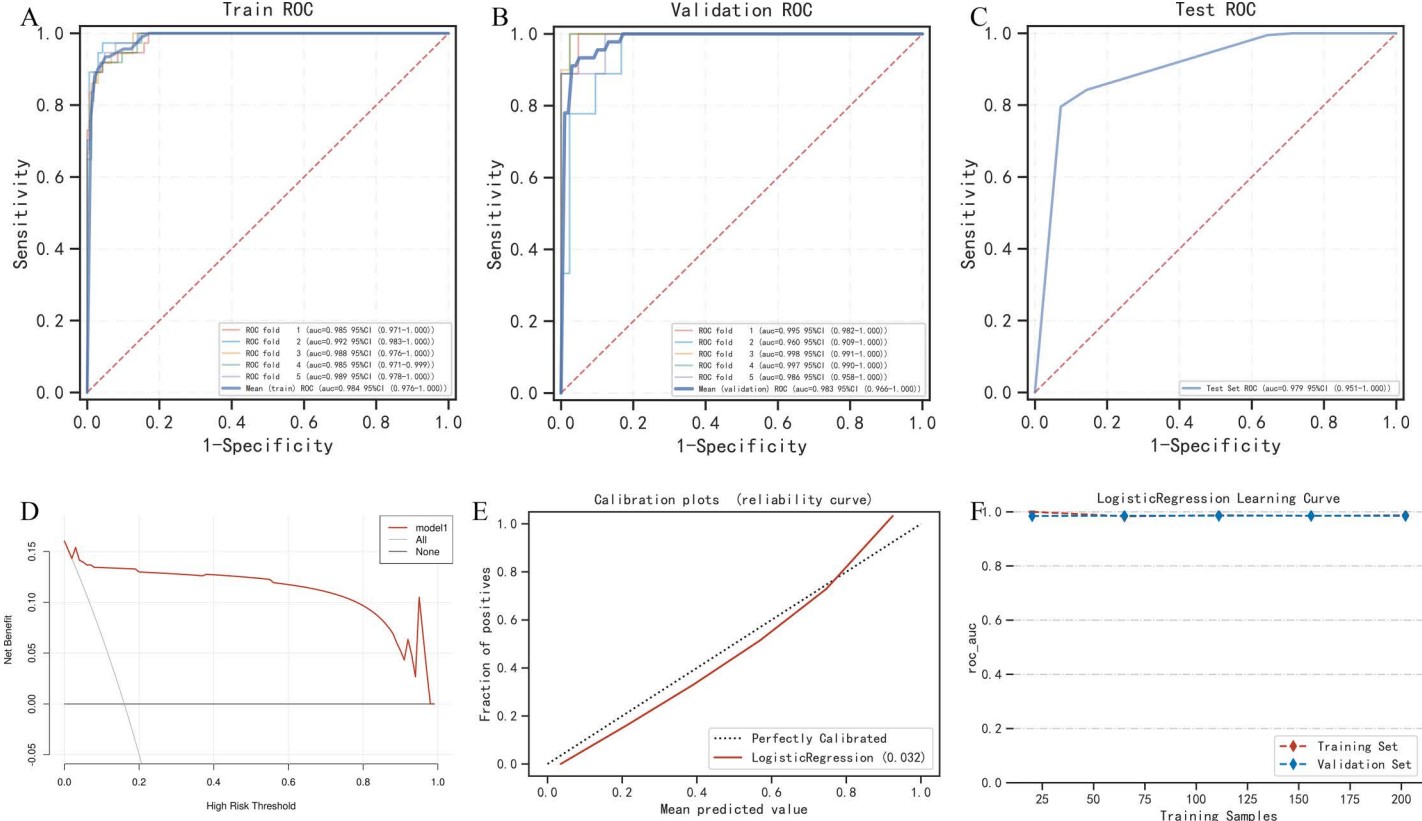

**Fig 5. The performance of the stacking machine learning model.** (**A**) Receiver operating characteristics (ROC) curves of 5-fold cross-validation in the training set. (**B**) ROC curves of 5-fold cross-validation in the validation set. (**C**) ROC curves in the testing set. (**D**) Decision curve analysis (DCA) of the stacking machine learning model. (**E**) Calibration plots of the stacking machine learning model. (**F**) Changes in areas under the ROC curves with the training samples in the training and validation sets.

present study, the potential ROS1 and ALK mutations might lead to a higher risk of VTE in non-smoking patients.

The EGFR exon 21 mutation was independently associated with VTE, but previous studies reported conflicting results. Indeed, Wang et al. [37] reported that the proportion of EGFR mutation in patients with VTE was higher than in the non-VTE group. Dou et al. [21] found that EGFR wild type was associated with an increased risk of VTE. A meta-analysis demonstrated that patients with EGFR did not show a significantly increased risk for VTE [20]. The present study found that the exon 19 mutation had a lower risk of VTE than the exon 21 mutation and had a similar risk to wild-type EGFR. Therefore, the EGFR exon 21 mutation was included in the machine learning model.

This study established a stacking machine learning model to develop an effective predictive model for VTE. The stacking machine learning model can improve the predictive power and blend a heterogeneous group of algorithms to expose distinct, complementary aspects of the data [24]. In this study, the LGBM Classifier, RandomForest Classifier, and GaussianNB were selected as the first layer of the stacking machine learning model, and logistic regression was chosen as the second layer meta-learning model. All the above machine learning models have been previously shown to be effective classification algorithms [38], and they demonstrated the most powerful predictive value among the nine machine learning models tested. In the present study, the stacking machine learning model showed a significant ability to predict

VTE. The risk of VTE increased with the score. The results also demonstrated that the model was stable and had a potential value to guide clinical practice. The model developed here fares better than previous machine learning models with an ROC AUC of 0.87 [39] and a C-index of 0.791-0.843 [40].

There were some limitations to this study. Firstly, there was no external validation, and it was a retrospective study. A prospective study would be an opportunity to validate the model established here. Secondly, the tumor specimens could not be tested for a large panel of gene mutations due to economic and ethical reasons. This study had to work with the mutations routinely tested in the clinical setting and could only include one driver mutation. Future prospective studies might be able to address that issue.

In conclusion, the present study identified six significant factors for VTE in patients with NSCLC undergoing resection. The EGFR exon 21 mutation was associated with VTE. Based on the six factors, the stacking machine learning model showed nearly perfect performance in predicting the risk of postoperative VTE in patients with lung adenocarcinoma. It had a potential value to guide the clinical practice in preventing postoperative VTE in patients with NSCLC.

## Supporting information

**STROBE Checklist.  This is the STROBE Checklist.**
(DOC)

## Acknowledgments

None.

## Author contributions

**Conceptualization:** Yonghui Di.

**Data curation:** Ligang Hao, Junjie Zhang.

**Formal analysis:** Ligang Hao, Junjie Zhang.

**Funding acquisition:** Ligang Hao, Junjie Zhang, Yonghui Di.

**Investigation:** Ligang Hao, Junjie Zhang, Yonghui Di, Zheng Qi, Peng Zhang.

**Methodology:** Ligang Hao.

**Project administration:** Ligang Hao, Junjie Zhang, Yonghui Di.

**Resources:** Ligang Hao, Junjie Zhang, Zheng Qi, Peng Zhang.

**Software:** Ligang Hao.

**Supervision:** Ligang Hao, Junjie Zhang, Zheng Qi.

**Validation:** Ligang Hao, Junjie Zhang, Yonghui Di, Peng Zhang.

**Visualization:** Ligang Hao, Junjie Zhang, Yonghui Di, Zheng Qi, Peng Zhang.

**Writing – original draft:** Ligang Hao, Junjie Zhang.

**Writing – review & editing:** Ligang Hao, Junjie Zhang.

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
