## [Decision Letter · Decision Letter 0]

10 Jan 2025

PONE-D-24-07620Predicting Postoperative Venous Thromboembolism in Non-Small Cell Lung Cancer: A Stacking Machine Learning ApproachPLOS ONE

Dear Dr. Di,

Thank you for submitting your manuscript to PLOS ONE. After careful consideration, we feel that it has merit but does not fully meet PLOS ONE’s publication criteria as it currently stands. Therefore, we invite you to submit a revised version of the manuscript that addresses the points raised during the review process.

**ACADEMIC EDITOR: **Overall this is well written manuscript, however reviewers raised multiple comments. As one of the reviewers pointed out please explain the high rate of VTE in this cohort of patients. Is it related to any other uncaptured risk factors, as authors couldn't assess other gene mutations. Please add other potential gene mutations in the discussion which increases the VTE risk in this cohort. Reiewer1 recommended rejection and reviewer2 recommended major revision. As an academic editor, I feel this paper can add value to the existing literature and potentially can use the described ML methodology in VTE risk stratification. Please address reviewer 1 and 2 comments for further consideration of the manuscript publication. ==============================

We look forward to receiving your revised manuscript.

Kind regards,

Madhuradhar Chegondi, MD

Academic Editor

PLOS ONE

Journal Requirements:

This work was supported by the Key Development Plan of XingTai (ZC20301 to Junjie Zhang and 2022ZC271 to Ligang Hao).

Reviewers' comments:

Reviewer's Responses to Questions

**Comments to the Author**

1. Is the manuscript technically sound, and do the data support the conclusions?

Reviewer #1: No

Reviewer #2: Yes

2. Has the statistical analysis been performed appropriately and rigorously? 

Reviewer #1: I Don't Know

Reviewer #2: Yes

3. Have the authors made all data underlying the findings in their manuscript fully available?

Reviewer #1: Yes

Reviewer #2: Yes

4. Is the manuscript presented in an intelligible fashion and written in standard English?

Reviewer #1: Yes

Reviewer #2: Yes

5. Review Comments to the Author

Reviewer #1: The authors report on predicting rates of VTE after thoracic surgery in patients with NSCLC. Issues with the paper include:

1. The VTE rate of 16% after surgery in a group of patients who received 4 wks of nadroparin prophylaxis is extraordinarily high. How do the authors account for such a high rate (even accounting for screening US and CT)?

2. The abstract, title etc should all mention that this is prediction of screen-detected VTE as a failure of thromboprophylaxis. This is not highlighted anywhere other than the Methods.

3. this is described as a machine learning paper, but it appears that EGFR mutation status was forced into the analysis. Is that correct? Or did the ML approach identify EGFR mutation status as a significant variable?

4. Extrapolation of these data to real world where screening is not typically done nor is 4 wks of thromboprophylaxis is difficult to ascertain.

Reviewer #2: INTRODUCTION: suggest to reduce to one page

METHODS: Line 113 to 114, please move to RESULTS portion.

RESULTS: Patient Characteristics, line 163. Please mention a few highlights of the overall patient characteristics. For table 1, Please adjust decimal as appropriate also: round percentage to nearest tenth and present ages as whole numbers.

DISCUSSION: Please summarize main findings of study into one paragraph.

Line 250. How do the results for d dimer fit into this paragraph?

Any comparative literature that uses machine learning as well that have been validated? How does your model compare to validated ones?

Please use past tense for completed actions and present tense for general validity.

6. PLOS authors have the option to publish the peer review history of their article (what does this mean? ). If published, this will include your full peer review and any attached files.

**Do you want your identity to be public for this peer review?** For information about this choice, including consent withdrawal, please see our Privacy Policy .

Reviewer #1: No

Reviewer #2: No

---

## [Author Response · Author response to Decision Letter 1]

8 Feb 2025

Dear Editor，

We are truly grateful to yours and other reviewers’ critical comments and thoughtful suggestions on our manuscript (Predicting A Failure of Postoperative Thromboprophylaxis in Non-Small Cell Lung Cancer: A Stacking Machine Learning Approach). Based on these comments and suggestions, we have made careful modifications on the original manuscript. We hope the new manuscript will meet your magazine’s standard. You will find our point-by-point responses to the reviewers’ comments/ questions as following.

Journal Requirements:

Thanks for the good advices from editors. We ensure that our manuscript meets PLOS ONE's style requirements, including those for file naming.

Thanks for the good advices from editors. I have Update my Information.

Thanks for the good advices from editors. We have furtherly developed model online to ensure that your code is shared in a way that follows best practice and facilitates reproducibility and reuse.

Please state what role the funders took in the study. If the funders had no role, please state: The funders had no role in study design, data collection and analysis, decision to publish, or preparation of the manuscript.""

Thanks for the good advices from editors. This work was supported by the Key Development Plan of XingTai (ZC20301 to Junjie Zhang and 2022ZC271 to Ligang Hao). And we have added the statement in our cover letter.

Thanks for the good advices from editors. we rechecked our manuscript that there was no ethics statement in other section.

Thanks for the good advices from editors. There was no Supporting Information file, and we state at the end of your manuscript.

Reviewers' comments:

Review Comments to the Author

Reviewer #1: The authors report on predicting rates of VTE after thoracic surgery in patients with NSCLC. Issues with the paper include:

1. The VTE rate of 16% after surgery in a group of patients who received 4 wks of nadroparin prophylaxis is extraordinarily high. How do the authors account for such a high rate (even accounting for screening US and CT)?

Thanks for the good advices from reviews. Previous study revealed that lung cancer is the most commonly identified malignancy in patients with VTE, with an incidence of 3%-13.9% in patients with VTE and 3.8% in patients with pulmonary embolism (PE). And incidence of 15.5% was reported in patients with lung resection, which was similar to our result. And much more stage III patients was enrolled in our study than previos study which also elevated the incidence of VTE.

2. The abstract, title etc should all mention that this is prediction of screen-detected VTE as a failure of thromboprophylaxis. This is not highlighted anywhere other than the Methods.

Thanks for the good advices from reviews. We mention that this is prediction of screen-detected VTE as a failure of thromboprophylaxis in title and abstract.

3. this is described as a machine learning paper, but it appears that EGFR mutation status was forced into the analysis. Is that correct? Or did the ML approach identify EGFR mutation status as a significant variable?

Thanks for the good advices from reviews. In our present study, EGFR mutation status was not forced into the analysis. It was identified as an independent risk factor by using univariable and multivariable analysis.

4. Extrapolation of these data to real world where screening is not typically done nor is 4 wks of thromboprophylaxis is difficult to ascertain.

Thanks for the good advices from reviews. We pay much more attention to thromboprophylaxis since 2018. Every patients underwent surgery in our department was followed up every week to ensure that the screening and thromboprophylaxis was done under as the guidance of us.

Reviewer #2:

INTRODUCTION: suggest to reduce to one page.

Thanks for the good advices from reviews.

METHODS: Line 113 to 114, please move to RESULTS portion.

Thanks for the good advices from reviews. we move it to RESULTS portion.

RESULTS: Patient Characteristics, line 163. Please mention a few highlights of the overall patient characteristics. For table 1, Please adjust decimal as appropriate also: round percentage to nearest tenth and present ages as whole numbers.

Thanks for the good advices from reviews. We mention a few highlights of the overall patient characteristics and adjust decimal round percentage to nearest tenth and present ages as whole numbers.

DISCUSSION: Please summarize main findings of study into one paragraph.

Thanks for the good advices from reviews. We summarize main findings of study into the last paragraph.

Line 250. How do the results for d dimer fit into this paragraph?

Thanks for the good advices from reviews. We recorrect that paragraph as following: In this study, the D-dimer levels were evaluated before surgery to reduce the influence of other factors as much as possible, so that it could reflect the hypercoagulable state of patients much more precisive. In this study, elevated D-dimer levels was also significantly related to VTE as an independent risk factor with OR=2.929 which was familiar to previous study.

Any comparative literature that uses machine learning as well that have been validated? How does your model compare to validated ones?

Please use past tense for completed actions and present tense for general validity.

Thanks for the good advices from reviews. It was demonstrated that The model developed here fares better than previous machine learning models with an ROC AUC of 0.87 [39] and a C-index of 0.791-0.843 [40] in our manuscript.

Thanks for the good advices from reviewers, we submitted it as a supplemental figure.

Thanks for addressing the comments!

We hope that these revisions are satisfactory and that the revised version will be acceptable for publication in PLOS ONE.

Thank you very much for your work concerning our paper.

Wish you all the best!

Sincerely yours

---

## [Editor Report · Decision Letter 1]

23 Feb 2025

Predicting A Failure of Postoperative Thromboprophylaxis in Non-Small Cell Lung Cancer: A Stacking Machine Learning Approach

PONE-D-24-07620R1

Dear Dr. Di

We’re pleased to inform you that your manuscript has been judged scientifically suitable for publication and will be formally accepted for publication once it meets all outstanding technical requirements.

Kind regards,

Madhuradhar Chegondi, MD

Academic Editor

PLOS ONE
---

## [Editor Report · Acceptance letter]

PONE-D-24-07620R1

PLOS ONE

Dear Dr. Di,

I'm pleased to inform you that your manuscript has been deemed suitable for publication in PLOS ONE. Congratulations! Your manuscript is now being handed over to our production team.

Kind regards,

on behalf of

Dr. Madhuradhar Chegondi

Academic Editor

PLOS ONE